# B-cell epitope discovery: The first protein flexibility-based algorithm–Zika virus conserved epitope demonstration

**Daniel W. Biner** [○], **Jason S. Grosch**[○], **Peter J. Ortoleva***

Department of Chemistry, Indiana University, Bloomington, Indiana, United States of America

○ These authors contributed equally to this work.
* ortoleva@indiana.edu

## Abstract

Antibody-antigen interaction–at antigenic local environments called B-cell epitopes–is a prominent mechanism for neutralization of infection. Effective mimicry, and display, of B-cell epitopes is key to vaccine design. Here, a physical approach is evaluated for the discovery of epitopes which evolve slowly over closely related pathogens (conserved epitopes). The approach is 1) protein flexibility-based and 2) demonstrated with clinically relevant enveloped viruses, simulated via molecular dynamics. The approach is validated against 1) seven structurally characterized enveloped virus epitopes which evolved the least (out of thirty-nine enveloped virus-antibody structures), 2) two structurally characterized non-enveloped virus epitopes which evolved slowly (out of eight non-enveloped virus-antibody structures), and 3) eight preexisting epitope and peptide discovery algorithms. Rationale for a new benchmarking scheme is presented. A data-driven epitope clustering algorithm is introduced. The prediction of five Zika virus epitopes (for future exploration on recombinant vaccine technologies) is demonstrated. For the first time, protein flexibility is shown to outperform solvent accessible surface area as an epitope discovery metric.

## Introduction

B-cell epitopes are localities of antigens targeted by the humoral immune response, via antibodies and B-cell receptors, to protect the extracellular space (e.g. the blood plasma) [1]. Molecules which effectively mimic B-cell epitopes are vaccines [2–4]. Consequently, structure-based B-cell epitope discovery has emerged as a promising foundational step in rational vaccine design [5,6]. Despite promise, 1) ambiguity surrounding the definition of an epitope, 2) limitations of current performance benchmarking approaches, 3) a scarcity of benchmark datasets, and 4) an overabundance of solvent accessible surface area-based metrics suggest there is ample room for improvement within the field. For example, while some enveloped virus [7] epitopes (like those of the Zika virus [8] (ZIKV)) [9–11] have been studied, a comprehensive, quantitative, uniform, and structure-based epitope analysis of a clinically relevant enveloped virus (like ZIKV) has yet to be explored. Likewise, 1) "cryptic" epitopes hidden within virus structures, [12] 2) pathogen morphological diversity, [13] 3) full pathogen

**Data Availability Statement:** All MD simulation files are available from the IUScholarWorks Repository (DOI: 10.5967/v844-r290).

**Funding:** PJO Grant #: UL1 TR001108 the National Institutes of Health, National Center for Advancing

Translational Sciences, Clinical and Translational Sciences Award https://ncats.nih.gov/ctsa NO Indiana University Grant #: CNS-0521433 National Science Foundation https://www.nsf.gov/ NO This research was supported in part by Lilly Endowment, Inc., through its support for the Indiana University Pervasive Technology Institute. NO This work was supported in part by Shared University Research grants from IBM, Inc., to Indiana University. NO.

**Competing interests:** The authors have declared that no competing interests exist.

structural dynamics, [14] and 4) possible links between protein flexibility and immunogenicity [15,16] suggest an overreliance on solvent accessible surface area-based epitope discovery metrics [5] may be a major oversight within the field, especially when it comes to highly flexible, clinically relevant antigens (like ZIKV).

Although challenges still exist, recent advancements within the field of all-atom molecular dynamics simulation (MD) have opened up new opportunities to enhance our atomic level understanding of protein flexibility (within the context of the immune response) [17]. Recent MD investigations into multi-million atom VLP systems (like HIV-1 [18], satellite tobacco mosaic virus (STMV) [19], and human papillomavirus (HPV) [16,20]) show quantification of full pathogen structural dynamics 1) is possible and 2) can provide new insights into antibody-antigen interaction physics.

In addition to the limitations (within the field) which have already been noted (e.g. a scarcity of flexibility-based epitope discovery metrics), structure-based methods for the discovery of conserved epitopes (epitopes which evolve the least across closely related pathogens) have yet to be developed. Because highly conserved epitopes (by definition) generalize across many pathogens, highly conserved epitopes are of especially high value as antigenic targets. Ineffectiveness of current vaccine technologies in eliciting antibodies able to bind a diverse set of closely related pathogens has been a challenge in vaccine design [21,22]. Multivalent, live-attenuated vaccines have been introduced to solve the complications mentioned above; however, simultaneous vaccine-based display of related pathogen epitopes has resulted in unbalanced prophylactic protection–suggesting new approaches are needed [23]. One promising, new approach is the presentation (or display) of conserved epitopes on recombinant vaccines [24].

Here, previous studies of *in silico* immunology [16,20] are extended by evaluating an algorithm for conserved epitope discovery, based on protein flexibility–as measured via root-mean-square fluctuation (RMSF) of isolated protein and virus-like particle (VLP) protein residues. The method is developed with flaviviruses simulated via MD. The approach is validated against 1) seven structurally characterized flavivirus epitopes (from thirty-nine flavivirus-antibody structures) with the lowest phylogeny-based evolutionary rates (described by Ashkenazy, H., et al. (2016) [25] as the rate at which a structurally aligned residue changes over a phylogenic tree of proteins with shared ancestry), 2) two structurally characterized human papillomavirus (HPV) epitopes (from eight HPV-antibody structures) with low phylogeny-based evolutionary rates, and 3) eight preexisting epitope and peptide discovery algorithms [5]. To enhance the epitope dataset (for the prediction of currently uncharacterized ZIKV epitopes), epitopes from seven flaviviruses are structurally aligned. Using 1) a new (data-driven) clustering algorithm, 2) a new epitope organizational model, 3) a new epitope discovery performance benchmarking scheme (which addresses bias in previous methods), and 4) a new epitope discovery benchmark dataset–all presented here for the first time–the prediction of five ZIKV epitopes (which provide starting points for future presentation on recombinant vaccine technologies) is demonstrated. Physical insights identified here 1) supply context for understanding (seemingly contradictory) previous reports on protein flexibility's role in the humoral immune response [15,16,20,26] and 2) shed new light on immunologically relevant distinctions between clinically relevant epitope subsets. Notably, for the first time [26,27], protein flexibility is shown to outperform solvent accessible surface area as an epitope discovery metric.

## Results/Discussion

### Epitope discovery performance benchmarking and rationale for a new method

Epitope discovery performance benchmarking on a compilation of all ZIKV-aligned epitopes and associations between isolated protein convex hull scores vs. other metrics are shown in

**Table 1. Epitope discovery performance benchmarking on a compilation of epitopes.**

| Metric | ROCAUC | ± | PRAUC | ± | Rho | p | Type |
|---|---|---|---|---|---|---|---|
| hvlp | 0.78 | 0.04 | 0.81 | 0.04 | 0.43 | < 0.01 | SASA |
| hmon | 0.75 | 0.04 | 0.77 | 0.04 | — | — | SASA |
| epro | 0.74 | 0.02 | 0.76 | 0.03 | 0.88 | < 0.01 | SASA |
| fmon* | 0.73 | 0.02 | 0.74 | 0.03 | 0.73 | < 0.01 | RMSF |
| dtope | 0.72 | 0.03 | 0.74 | 0.03 | 0.61 | < 0.01 | SASA |
| opia | 0.69 | 0.07 | 0.70 | 0.06 | 0.61 | < 0.01 | SASA |
| bpro | 0.66 | 0.02 | 0.71 | 0.06 | 0.51 | < 0.01 | SASA |
| fmon | 0.66 | 0.02 | 0.71 | 0.03 | 0.48 | < 0.01 | RMSF |
| tfac | 0.63 | 0.05 | 0.65 | 0.04 | 0.59 | < 0.01 | RMSF |
| fvlp | 0.61 | 0.04 | 0.66 | 0.01 | 0.45 | < 0.01 | RMSF |
| iupred | 0.57 | 0.09 | 0.63 | 0.07 | 0.08 | 0.12 | SEQ |
| ppisp | 0.56 | 0.11 | 0.56 | 0.08 | 0.11 | 0.03 | SASA |

Metrics are ordered from the top to bottom in terms of highest ROCAUC and PRAUC product. Spearman rho (r) and p-values are shown for associations between isolated protein convex hull scores (hmon) vs. VLP protein convex hull scores (hvlp), partially isolated protein RMSF (fmon*), DiscoTope scores (dtope), Epitopia scores (opia), ElliPro scores (epro), BEpro scores (bpro), isolated protein RMSF (fmon), VLP protein RMSF (fvlp), temperature factors (tfac), cons-PPISP scores (ppisp), and IUPred scores (iupred). The primary type of structural information utilized for each metric is shown under the heading titled Type (solvent accessible surface area (SASA), protein flexibility (RMSF), and sequence information (SEQ)).

Table 1. The raw ZIKV-aligned epitope dataset is provided in S1 Dataset. For comparison, benchmarking on individual ZIKV-aligned epitopes is shown in S1 Table.

For epitope discovery performance benchmarking, the value of no discrimination [28] is equivalent to the Area Under the Curve (AUC) value associated with performance no better than chance (as discussed in the *Epitope discovery performance benchmarking and rationale for a new method* section of the *Methods*). Because the value of no discrimination assists comparison between Precision-Recall vs. Receiver Operating Characteristic analysis, it is provided for each AUC measurement discussed below.

Comparison of new epitope discovery performance indicator, Precision-Recall Area Under the Curve (PRAUC), with preexisting performance indicator, Receiver Operating Characteristic Area Under the Curve (ROCAUC), shows ROCAUC can overestimate epitope discovery performance when epitopes are analyzed individually, as expected. Over all metrics, the mean ROCAUC was $0.13 \pm 0.07$ (above the value of no discrimination, 0.50) [28] and the mean PRAUC was $0.07 \pm 0.05$ (above each value of no discrimination, a mean of $0.05 \pm 0.02$) when benchmarking was performed on individual epitopes 1) supporting the idea epitopes are much smaller than the entirety of an antigen (comprising only ~5% of the number of antigen residues here, 402) and 2) telling us, despite strong capacity to rank non-epitope residues below epitope residues (a high ROCAUC mean), the average metric performed poorly at pinpointing individual epitopes from other known epitopes (a low PRAUC mean) (S1 Table) [29]. For comparison, over all metrics, the mean ROCAUC was $0.17 \pm 0.07$ (above the value of no discrimination, 0.50) and mean PRAUC was $0.16 \pm 0.07$ (above the value of no discrimination, 0.54) when benchmarking was performed on a compilation of epitopes telling us 1) the proportion of epitope residues and non-epitope residues had parity when epitopes were compiled (similar ROCAUC and PRAUC values of no discrimination, 0.50 and 0.54 respectively) and 2) the average metric had a higher capacity to pinpoint a compilation of epitopes (a high PRAUC mean) than individual epitopes ($t(22) = 3.77$ p = < 0.01) (Table 1). These results make sense as structural metrics are associated with the entirety of a protein antigen and antigens contain multiple epitopes [30]. These results are also important for interpreting previous epitope

discovery benchmarking attempts [5]. From this point forward, all epitope discovery performance benchmarking discussion will refer to results obtained using the compilation of epitopes benchmarking scheme.

As a note going forward, because a single flavivirus gene product forms the basis for epitope discovery performance benchmarking here–the envelope protein [31], we refer to this protein as the ZIKV (or flavivirus) protein (for more detail, see the *Model preparation and molecular dynamics simulations* section of the *Methods*). Additionally, a ZIKV VLP protein refers to the envelope proteins which comprise the ZIKV VLP model examined here, a hollow protein cage, as shown in S3 Fig.

Epitope discovery performance benchmarking comparison shows solvent accessible surface area-based metrics (layered convex hulls, ElliPro [32], DiscoTope [33], BEpro [34], Epitopia [35], and cons-PPISP [36]) performed both the strongest and weakest of all metrics against all thirty-nine ZIKV-aligned flavivirus epitopes (Table 1). More specifically, VLP protein convex hull scores performed with one of the strongest capacities to rank epitope residues above non-epitope residues (the highest ROCAUC mean) and to pinpoint epitope residues (one of the highest PRAUC means) likely because 1) flavivirus structures mature as they are secreted from the cell into the extracellular space [37], 2) the mature ZIKV structure was used to model the VLP examined here, [8] and 3) the humoral immune response protects the extracellular space from foreign objects via B-cell antibodies [38]. For comparison, a moderate Spearman correlation and overlapping performance were observed between isolated and VLP protein convex hull scores (r = 0.43, p < 0.01, n = 402) perhaps because 1) the transmembrane protein region protrudes from the isolated protein, enhancing convex hull scores of residues which are also exposed on the VLP [8], or 2) the extracellular space sees a diverse ensemble of flavivirus morphologies (outside the mature cryo-EM reconstruction), as supported by recent studies [12–14]. Overall, these results 1) join growing support for the hypothesis accessibility to antibody binding is one prerequisite for an epitope's existence [5,33,39] and 2) suggest morphological diversity may challenge our assumptions as to which regions of a pathogen are accessible to antibody binding.

Shifting focus to protein flexibility-base metrics, examination of epitope discovery performance of isolated protein flexibility and partially isolated protein flexibility (incorporating the transmembrane region and two bound M proteins; labeled fmon* in Table 2) reveals partially isolated protein flexibility performed 1) on par with ElliPro and

**Table 2. Conserved epitope discovery performance benchmarking.**

| Metric | ROCAUC | ± | PRAUC | ± | Rho | p | Type |
|---|---|---|---|---|---|---|---|
| fmon-fvlp | 0.82 | 0.08 | 0.46 | 0.08 | 0.53 | < 0.01 | RMSF |
| fmon | 0.72 | 0.07 | 0.48 | 0.10 | — | — | RMSF |
| fmon* | 0.71 | 0.07 | 0.46 | 0.13 | 0.72 | < 0.01 | RMSF |
| epro | 0.69 | 0.07 | 0.45 | 0.10 | 0.53 | < 0.01 | SASA |
| opia | 0.64 | 0.03 | 0.27 | 0.05 | 0.53 | < 0.01 | SASA |
| hmon | 0.62 | 0.03 | 0.24 | 0.07 | 0.48 | < 0.01 | SASA |
| ppisp | 0.69 | 0.12 | 0.18 | 0.14 | 0.31 | < 0.01 | SASA |
| fvlp | 0.59 | 0.07 | 0.14 | 0.03 | 0.76 | < 0.01 | RMSF |
| dtope | 0.62 | 0.09 | 0.13 | 0.04 | 0.65 | < 0.01 | SASA |
| bpro | 0.58 | 0.05 | 0.13 | 0.05 | 0.58 | < 0.01 | SASA |
| tfac | 0.56 | 0.11 | 0.09 | 0.02 | 0.37 | < 0.01 | RMSF |
| iupred | 0.50 | 0.04 | 0.07 | 0.01 | 0.04 | 0.44 | SEQ |
| hvlp | 0.39 | 0.10 | 0.06 | 0.01 | 0.12 | 0.02 | SASA |

DiscoTope (a PRAUC mean of 0.73 ± 0.02 and ROCAUC mean of 0.74 ± 0.03) and 2) significantly stronger than isolated protein flexibility at ranking ZIKV-aligned epitope residues above all other residues (a higher mean ROCAUC) (t(8) = 3.88 p = < 0.01). A stronger correlation was observed between partially isolated protein flexibility and isolated protein convex hull scores (r = 0.73, p < 0.01, n = 402) than isolated protein flexibility and isolated protein convex hull scores (r = 0.48, p < 0.01, n = 402) further highlighting differences between isolated and partially isolated protein flexibility. Together, these results signify 1) protein flexibility can perform with relatively high capacity to pinpoint epitopes and 2) protein flexibility metrics can have a range of similarities with solvent accessible surface area depending on protein local environment.

Unlike partially isolated protein flexibility, other flexibility metrics examined here (isolated protein flexibility, VLP protein flexibility, and temperature factors) 1) were moderately correlated with and 2) performed below most solvent accessible surface area-based metrics (against all thirty-nine ZIKV-aligned flavivirus epitopes) (Table 1). Despite relatively weak performance (in comparison with solvent accessible surface area-based metrics and partially isolated protein flexibility), these flexibility-based metrics retained some capacity to pinpoint ZIKV-aligned flavivirus epitope residues (PRAUC means above the value of no discrimination, 0.54). RMSD equilibration plots indicate, despite performance overlap, differences between isolated and VLP protein flexibility were likely associated with steric hindrance between adjacent VLP proteins, a 0.15 Å RMSD standard deviation (on average) for all VLP proteins (analyzed individually) vs. a 0.37 Å RMSD standard deviation for the isolated protein (over the last 20 ns; Fig 1). These results reveal 1) isolated protein flexibility was dampened upon incorporation into a VLP and 2) pathogen morphological diversity may complicate our understanding of protein flexibility's role within the humoral immune response.

To ensure the ZIKV VLP was equilibrated prior to VLP protein flexibility calculation, the VLP waters exchange rates were quantified. Over the last 20 ns of the simulation, the rate of waters entering and exiting the ZIKV VLP were 122,973 ± 1.344 and 123,105 ± 1,328 per ns (~5% of the total waters in the capsid), respectively. These rates are approximately two orders of magnitude higher than those observed for a simulation of the mature HIV capsid [18], highlighting the permeability of the ZIKV VLP. Relatively high permeability of the ZIKV VLP (compare to the mature HIV capsid) could be related to 1) differences between envelope proteins and capsid proteins and 2) the absence of membrane and membrane-associated protein components within the ZIKV VLP. Overall, these results show the internal pressure of the VLP was equilibrated prior to RMSF calculation.

Although protein flexibility-based metrics (which were most distinct from solvent accessible surface area-based metrics here) failed to outperform solvent accessible surface area-based metrics (in terms of epitope discovery performance against all ZIKV-aligned epitopes), 1) very strong correlations between isolated protein flexibility of seven structurally related flaviviruses (a mean Spearman r of 0.80 ± 0.04 and p < 0.01 over flavivirus structures; S1 Fig) and 2) seemingly contradictory conclusions from previous studies (examining flexibility's role in the humoral immune response) [15,16,20] suggest an external factor such as antibody flexibility (as hypothesized here) may influence the relationship between protein flexibility and epitopes [40,41].

To understand how consideration of protein flexibility differences across antibodies [40,41] could reveal a stronger association between protein flexibility-based metrics (which were most distinct from solvent accessible surface area-based metrics here) and epitopes, one must examine the combined influence of antibody flexibility and epitope flexibility on antibody-antigen binding affinity–as epitopes are, by definition, antigenic local environments bound by antibodies. First, one way of obtaining high antibody-antigen binding affinity is through

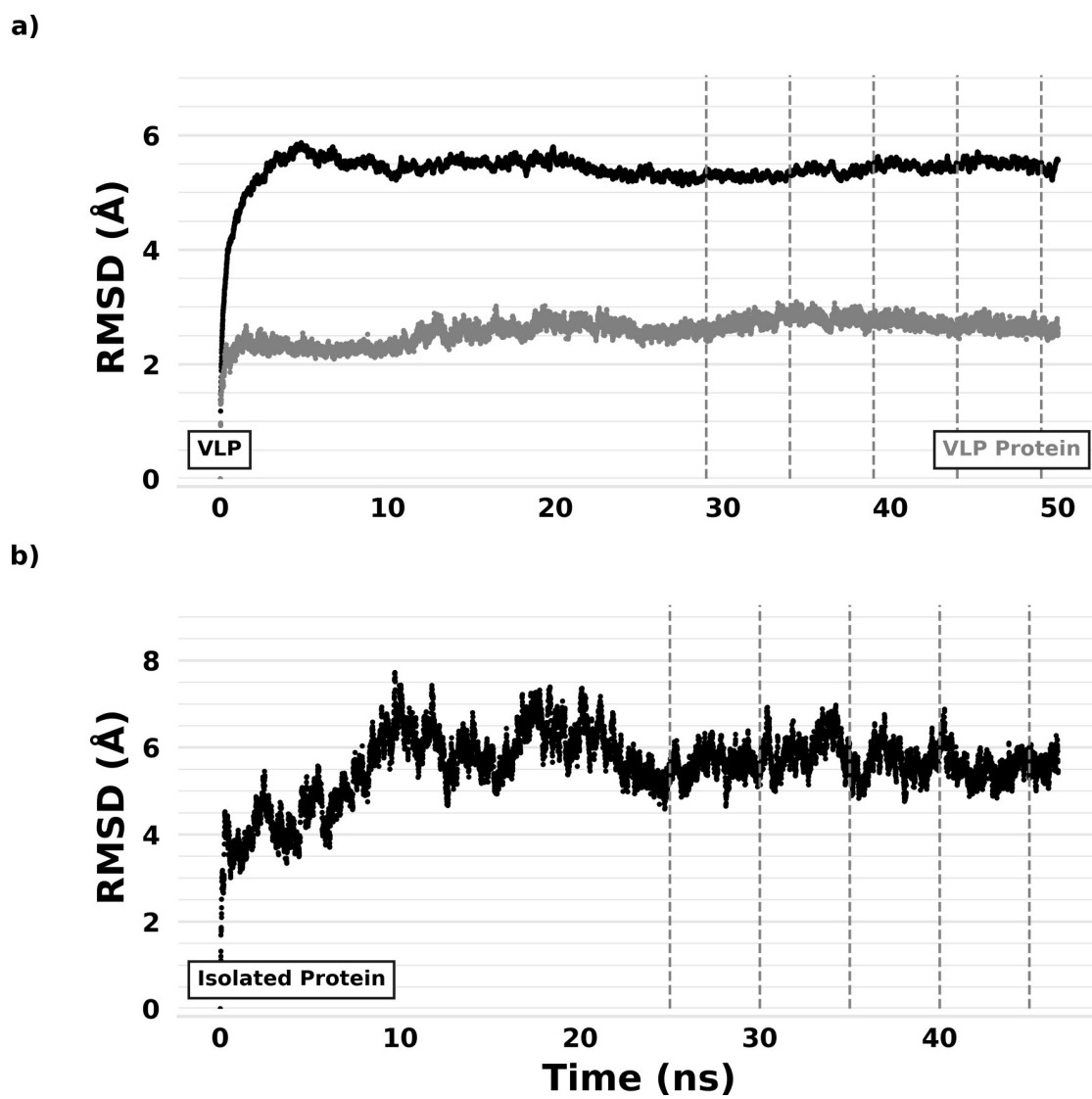

**Fig 1. ZIKV structural equilibration at 300 K.** a) RMSD from starting configuration for a) the ZIKV VLP (black trace), a single VLP protein (gray trace), and b) the isolated ZIKV protein are shown. Vertical lines represent trajectory bounds used for RMSF calculation.

maximization of contact between an antibody and an antigen. Second, to maximize contact between an antibody and an antigen, either the antibody or the antigen (more specifically the partner epitope on the antigen) must have flexibility. For example, when an antibody and an epitope are both rigid, contact is likely limited via steric hinderance, and, when an antibody and an epitope are both flexible, contact is likely limited via a high conformational entropy barrier. Using this logic, the relationship between protein flexibility and epitopes should be enhanced for epitopes bound by antibodies with relatively high rigidity compared with all antibodies. Broadly neutralizing antibodies, for example, have recently been shown to affinity mature through a process which rigidifies the antibody, while, at the same time, enhancing antibody-antigen binding specificity [40,41] suggesting affinity matured, broad-spectrum antibodies may preferentially target flexible regions of pathogens.

## Conserved epitope discovery performance benchmarking and the flexibility-based model

In light of results here and previous suggestions of flexibility's influence on broadly neutralizing antibody affinity maturation [40,41], epitope discovery performance benchmarking on a compilation of the top seven most conserved flavivirus epitopes (an epitope subset which 1) evolves slowly over structurally related pathogens and 2) is likely targeted by broad-spectrum antibodies) was examined (Table 2).

Metrics are ordered from the top to bottom in terms of highest ROCAUC and PRAUC product calculated against the top seven most conserved ZIKV-aligned epitopes. Spearman rho (r) and p-values are shown for associations between isolated protein RMSF (fmon) vs. (fmon-fvlp) the linear difference of isolated and VLP RMSF, partially isolated protein RMSF (fmon*), ElliPro scores (epro), Epitopia scores (opia), isolated protein convex hull scores (hmon), cons-PPISP scores (ppisp), BEpro scores (bpro), (hmon-hvlp) the linear difference of isolated and VLP convex hull scores, VLP protein RMSF (fvlp), DiscoTope scores (dtope), IUPred scores (iupred), temperature factors (tfac), and VLP protein convex hull scores (hvlp). The primary type of structural information utilized for each metric is shown under the heading titled Type (solvent accessible surface area (SASA), protein flexibility (RMSF), and sequence information (SEQ)).

Epitope discovery performance benchmarking comparison shows isolated protein flexibility performed above all other lone flexibility-based metrics (VLP protein RMSF and temperature factors) and all solvent accessible surface area-based metrics (layered convex hulls, ElliPro, DiscoTope, BEpro, Epitopia, and cons-PPISP) against the top seven most conserved flavivirus epitopes–the first report of a flexibility-based epitope discovery metric outperforming solvent accessible surface area-based metrics (Table 2) [5]. Examination of epitope discovery performance against the top two most conserved flavivirus epitopes shows isolated protein flexibility also maintained almost perfect capacity to 1) pinpoint residues from the top two conserved epitopes (a PRAUC mean of 0.98 ± 0.02; a value of no discrimination of 0.04) and 2) rank residues from the top two conserved epitopes above all other residues (a ROCAUC mean of 0.86 ± 0.13; a value of no discrimination of 0.50). Closer inspection reveals 1) isolated protein flexibility of seven structurally related flaviviruses retained comparable capacity to pinpoint residues from the top seven conserved epitopes (a mean PRAUC over flavivirus structures of 0.47 ± 0.03; a mean value of no discrimination of 0.08; S2 Table), 2) isolated protein flexibility from a ZIKV protein simulation without disulfides performed with a slightly lower capacity (despite overlapping capacity) to rank residues from the top seven conserved epitopes above all other residues (ROCAUC mean of 0.68 ± 0.07; a value of no discrimination of 0.50) than is isolated protein flexibility with disulfides, supporting separate studies suggesting disulfides play a structural role in flavivirus immunogenicity [42–46], 3) IUPred, a sequence-based measure of intrinsically disordered protein regions (which one might associate with protein flexibility) [47,48], performed no better than chance against the top seven conserved epitopes (ROCAUC and PRAUC means near their respective values of no discrimination, 0.50 and 0.08), and 4) ElliPro had overlapping performance with isolated protein flexibility in terms of capacity to pinpoint residues from the top seven conserved epitopes (despite a lower PRAUC mean).

Strong conserved epitope discovery performance with isolated protein flexibility comes as little surprise, as 1) conserved protein regions are necessary for successful flavivirus infection [49] and 2) protein flexibility (linked with major morphological shifts) is also necessary for successful flavivirus infection. For example, conformational dynamics are thought to facilitate flavivirus endosomal escape, and antibodies which block or alter these dynamics have been shown to neutralize infection [50].

Despite performance overlap and compared with isolated protein flexibility alone, VLP protein flexibility-based metrics (VLP protein RMSF and temperature factors) performed weakly at pinpointing conserved epitope residues (PRAUC means near the value of no discrimination, 0.08). However, when combined, the linear difference of isolated and VLP protein flexibility (using weights which minimized the root-mean-square difference between the two flexibility profiles) performed with stronger capacity to rank conserved epitope residues above all other residues (a distinctly higher ROCAUC distribution) compared with ElliPro (t(8) = 2.43, p = 0.04) (Table 2).

Examination of differences between isolated and partially isolated ZIKV protein flexibility reveals the two metrics performed similarly. However, unlike isolated protein flexibility, linear combination of partially isolated and VLP protein flexibility (using weights optimized via test and train dataset splits) resulted in conserved epitope discovery performance indistinguishable from ElliPro (a PRAUC mean of 0.49 ± 0.06 and ROCAUC mean of 0.75 ± 0.06). These results imply rotational and translation freedom of the fully isolated ZIKV protein, dampened upon assembly into the mature ZIKV virus structure, was one facet of the linearly combined, flexibility model's capacity to rank conserved epitope residues above all other residues. Like previously mentioned, broadly neutralizing flavivirus antibodies have been shown to mute or disrupt the dynamics of mature flavivirus structures, obstructing conformational motions required for endosomal escape and successful infection [50].

Visualization of the linearly combined, flexibility model on the ZIKV cryo-EM reconstruction [8] highlights a radial spiral in metric amplitude originating at the five-fold rotational icosahedral symmetry axes (Fig 2), a local environment thought to hold stress [51] of icosahedral virus curvature. Visual analysis indicates conserved epitopes may be important for 1) stabilizing the mature flavivirus morphology before endosomal escape and 2) destabilizing the mature flavivirus morphology for endosomal escape, as part of a morphological transition state. This makes sense as isolated ZIKV protein flexibility (isolated from other envelope proteins, M proteins, and the transmembrane region) is likely, at least somewhat, indicative of the flexibility sampled during flavivirus morphological transitions (e.g. immature to mature or mature to endosomal escape morphologies).

Inspection of the weights which optimized performance for the linearly combined, flexibility model using 1) test and train dataset splits and 2) root-mean-square difference minimization between the two individual flexibility metrics shows the two linear combination methods produced similar results. For example, weights which minimized the root-mean-square difference between isolated and VLP protein flexibility (0.378 and 0.622, respectively) were almost

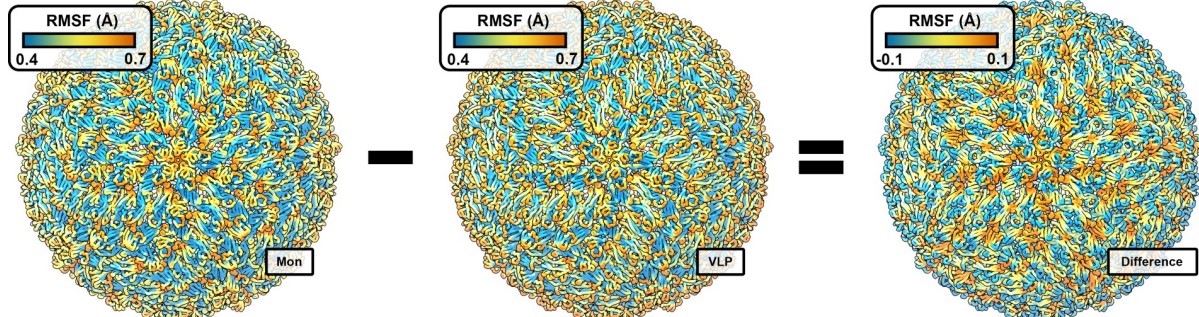

**Fig 2. Visualizing the linearly combined, flexibility model via the mature ZIKV cryo-EM reconstruction.** From left to right: Weighted isolated protein flexibility, weighted VLP protein flexibility, and the linearly combined, flexibility model are depicted over the mature ZIKV cryo-EM reconstruction (PDB ID: 5IRE) [8] (scales shown). VLP structures are oriented with the five-fold rotational icosahedral symmetry axis coming out of the figure plane.

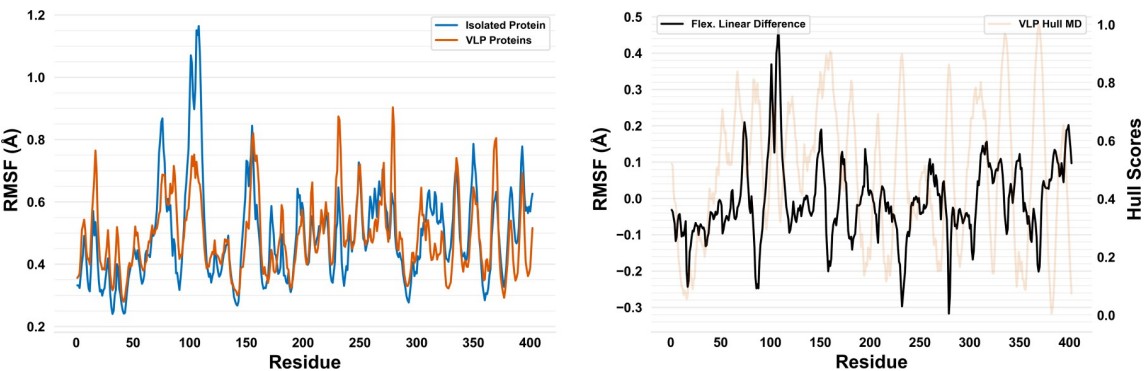

**Fig 3. Visualizing the linearly combined, flexibility model via RMSF profiles.** On the left, plots of the weighted isolated (blue trace) and VLP (red) protein flexibility metrics vs. residue number are overlaid. On the right, a plot of the linearly combined, flexibility model (black trace; using weights which minimized the root-mean-square difference between the two flexibility-based metrics) vs. residue number is shown. For comparison, on the right, a plot of VLP hull scores (red trace) is shown behind a plot of the linearly combined, flexibility model.

identical to weights which optimized performance for the linear combination (0.325 and 0.675, respectively). These results suggest the linearly combined, flexibility model was representative of the relative flexibility difference between the isolated ZIKV protein and ZIKV VLP proteins after baseline flexibility differences were accounted for (Fig 3).

For comparison, solvent accessible surface area-based metrics performed with relatively weak capacity to rank conserved flavivirus epitope residues above all other residues (relatively low ROCAUC means) and to pinpoint conserved epitope residues (relatively low PRAUC means) (Table 2). In fact, although isolated ZIKV protein convex hull scores had some capacity to rank conserved epitope residues above all other residues (a ROCAUC mean ~0.10 above the 0.50 value of no discrimination), ZIKV VLP protein convex hull scores ranked all other residues above conserved flavivirus epitope residues (a ROCAUC mean ~0.10 below the 0.50 value of no discrimination). Additionally, in contrast with protein flexibility, the linear difference of ZIKV isolated and VLP protein convex hull scores (using weights optimized via test and train dataset splits) performed with 1) a higher capacity to rank conserved flavivirus epitope residues above all other ZIKV residues (a higher ROCAUC mean, $0.71 \pm 0.04$; a value of no discrimination of 0.50) but 2) a much weaker capacity to pinpoint conserved flavivirus epitope residues (a lower PRAUC mean, $0.14 \pm 0.02$; a value of no discrimination of 0.08) compared with isolated ZIKV protein convex hull scores alone. Interestingly, cons-PPISP, a metric calibrated to typical protein-protein interfaces, performed with a stronger capacity to pinpoint epitope residues against the top seven most conserved flavivirus epitopes (a PRAUC mean ~0.10 higher than the 0.08 value of no discrimination) than against all thirty-nine flavivirus epitopes (a PRAUC mean around the 0.54 value of no discrimination). These results signify 1) protein-protein interaction sites play a possibly limited role in the existence of conserved epitopes, 2) differences between ZIKV isolated and VLP protein convex hull scores were accentuated via performance benchmarking against conserved flavivirus epitopes, and, relatedly, 3) it is evolutionarily advantageous to partially or transiently conceal conserved epitopes for immune evasion [49]. Interestingly, despite potential partial or transient concealment, engineered presentation of conserved epitopes on recombinant vaccines can still induce broadly reactive humoral immune responses [24]–highlighting the potential significance of 1) conserved epitope discovery methods in vaccine design and 2) engineered presentation of conserved epitopes on vaccine, protein scaffolds.

Shifting focus to the generalizability of the protein flexibility-based epitope discovery method presented here, performance reliability shows isolated flavivirus protein flexibility pinpoints conserved flavivirus epitope residues well across seven structures (a mean PRAUC of $0.47 \pm 0.03$ over flavivirus structures; S2 Table). Interestingly, isolated flavivirus protein flexibility appears to be conserved across flavivirus envelope proteins (a mean Spearman r of $0.80 \pm 0.04$ and $p < 0.01$ over flavivirus structures; S1 Fig)–a finding which may account for strong flexibility-based conserved epitope discovery performance here. The flavivirus NS3 protein was also recently shown to have a flexibility profile conserved across flaviviruses [52] despite a lack of sequence similarity.

Results here, along with the discovery of an ancestral protein linked with class II fusion proteins (e.g. the flavivirus envelope protein) [53], suggest the flexibility-based method presented here likely generalizes across pathogens with class II fusion proteins. Previous observations of protein flexibility's role in broadly neutralizing HIV antibody maturation [41] and the first principles nature of protein flexibility-based (as discussed in *Epitope discovery performance benchmarking and rationale for a new method* section of the *Results*) suggest the protein flexibility-based epitope discovery method could generalizes beyond pathogens with class II fusion proteins.

To better understand the generality of flexibility-based method presented here, protein flexibility's influence on conserved human papillomavirus [16] (HPV) epitopes was examined. Benchmarking was performed against two highly conserved (mean residue evolutionary rate of 1.52) [25] HPV epitopes (PDB IDs: 7CN2 [54], 3J8Z [55]) out of eight structurally characterized HPV epitopes (evolutionary rate range of 1.18–2.47). Like flaviviruses, linear combination of HPV isolated and VLP protein flexibility (using weights which minimized the root-mean-square difference between the two flexibility profiles) performed with the strongest capacity (of all metrics examined) to pinpoint conserved HPV epitope residues (a PRAUC mean 0.21 above the value of no discrimination, 0.07) (Table 3). For comparison, ElliPro and IUPred performed with some capacity to pinpoint conserved HPV epitopes (PRAUC means ~0.10 above the value of no discrimination, 0.07), while DiscoTope, BEpro, and cons-PPISP perform no better than chance at pinpointing conserved HPV epitopes (PRAUC means around the value of no discrimination, 0.07). These results show the physics-based protein flexibility model identified here does indeed generalize beyond clinically relevant enveloped viruses to clinically relevant non-enveloped viruses.

## Conserved epitope organization and epitope discovery demonstration

Of the seven highly conserved epitopes, 14% were associated with ZIKV, 0% were associated with dengue virus serotype 1, 14% were associated with dengue virus serotype 2, 14% were associated with dengue virus serotype 3, 29% were associated with dengue virus serotype 4,

**Table 3. Generalizability of the flexibility-based, conserved epitope model.**

|        | fmon-fvlp | epro | iupred | dtope | bpro | ppisp |
|--------|-----------|------|--------|-------|------|-------|
| **ROCAUC** | 0.81±0.06 | 0.82±0.06 | 0.73±0.06 | 0.67±0.07 | 0.63±0.08 | 0.44±0.07 |
| **PRAUC** | 0.28±0.06 | 0.18±0.07 | 0.15±0.07 | 0.09±0.02 | 0.09±0.02 | 0.06±0.01 |
| **Type** | RMSF | SASA | SEQ | SASA | SASA | SASA |

ROCAUC and PRAUC values calculated against two highly conserved HPV epitopes [54,55] are shown for the linear difference of isolated and VLP RMSF (fmon-fvlp), ElliPro scores (epro), IUPred scores (iupred), DiscoTope scores (dtope), BEpro scores (bpro), cons-PPISP scores (ppisp). The primary type of structural information utilized for each metric is shown under the heading titled Type (solvent accessible surface area (SASA), protein flexibility (RMSF), and sequence information (SEQ)).

**Table 4. Flavivirus epitope mean evolutionary rates.**

| PDB | ER | Flav. | PDB | ER | Flav. | PDB | ER | Flav. | PDB | ER | Flav. |
|---|---|---|---|---|---|---|---|---|---|---|---|
| 3I50 | -0.95 | W | 4UTB | -0.13 | D2 | 3UZV | 0.42 | D2 | 5LBS | 0.69 | Z |
| 5JHL | -0.93 | Z | 4UTA | -0.1 | D1 | 3UYP | 0.45 | D4 | 1ZTX | 0.73 | W |
| 3IXX | -0.91 | W | 5KVD | -0.04 | Z | 2R69 | 0.48 | D2 | 5VIC | 0.75 | D1 |
| 4BZ1 | -0.78 | D4 | 4UT9 | 0.03 | D2 | 5GZO | 0.52 | Z | 3IYW | 0.77 | W |
| 4AM0 | -0.7 | D4 | 3UZQ | 0.33 | D1 | 5KVE | 0.54 | Z | 5YWP | 0.78 | J |
| 4ALA | -0.63 | D3 | 4FFZ | 0.34 | D1 | 3UAJ | 0.54 | D4 | 5VIG | 0.86 | Z |
| 3IXY | -0.62 | D2 | 4FFY | 0.36 | D1 | 4L5F | 0.57 | D1 | 5Y0A | 0.86 | Z |
| 4AL8 | -0.51 | D1 | 5AAW | 0.36 | D4 | 5KVG | 0.62 | Z | 5YWO | 0.86 | J |
| 5LCV | -0.36 | Z | 3UZE | 0.38 | D3 | 5KVF | 0.63 | Z | 5GZN | 1.41 | Z |
| 4UT6 | -0.15 | D2 | 5H37 | 0.41 | Z | 5UHY | 0.64 | Z | — | — | — |

Flavivirus-antibody structures from which conserved epitopes were identified are ordered from the top to bottom and left to right in terms of mean residue evolutionary rate (ER) [25]. The PDB ID and flavivirus of the associated structures are shown (ZIKV (Z), dengue virus serotype 1–4 (D1-4), West Nile virus (W), Japanese encephalitis virus (J)).

29% were associated with West Nile virus, and 0% were associated with Japanese encephalitis virus (Table 4).

Examination of ZIKV-aligned epitope organization shows conserved flavivirus epitopes fall on the lower end of the epitope distribution, in terms of epitope residue quantity and epitope discontinuity. Conserved flavivirus epitopes had a mean of 8 ± 4 total residues and 3 ± 1 subsequences with 4 ± 3 residues. All thirty-nine ZIKV-aligned flavivirus epitopes had a mean of 18 ± 9 total residues and 5 ± 3 subsequences with 4 ± 4 residues. For comparison, a previous analysis of seventy-six antibody-antigen structures [56] found epitopes had a mean of 16 ± 3 total residues and 5 ± 2 subsequences with 2 ± 2 residues. These results further support the idea partial or transient concealment of conserved epitopes is evolutionarily advantageous [49], emphasizing the potential value of 1) conserved epitopes and 2) conserved epitope discovery methods in vaccine design.

Prediction of five ZIKV epitopes with 8 ± 7 total residues and 1 ± 0 subsequence of 5 ± 4 residues was demonstrated (Table 5). Visual inspection of predicted epitopes on the ZIKV cryo-EM reconstruction shows a spiral pattern focused around the five-fold rotational icosahedral symmetry axes (Fig 4). High continuity of predicted epitopes is likely a result of 1) the structure examined, 2) the parameters set within the epitope residue clustering algorithm (described below), and 3) the residue profile of the metric used for epitope prediction. The isolated and VLP protein flexibility linear difference threshold which minimized the root-mean-square difference between known conserved epitope organization and predicted conserved epitope organization was 0.10 Å (Fig 3). The highest performing epitope clustering residue-residue distance cutoff which minimized the root-mean-square difference between known conserved epitope organization and predicted conserved epitope organization was 6 Å.

**Table 5. Predicted conserved epitopes.**

| Predicted Epitope | ER | Predicted Epitope | ER | Predicted Epitope | ER |
|---|---|---|---|---|---|
| [147, 148, 149, 150, 151, 152]; 'QHSGMI' | -.94 | [195, 196]; 'GL' | 0.52 | [72, 73, 74, 75, 76, 97, 98, 99, 100, 101, 102, 103, 104, 105, 106, 107, 108, 109, 110, 111, 112]; 'SRCPTVDRGWGNGCGLFGKGS' | 1.74 |
| [171, 172]; 'PN' | 0.46 | [311, 312, 313, 314, 315, 316, 317, 318, 331]; 'AFTFTKIPQ' | 0.57 | | |

Predicted ZIKV epitopes are shown and ordered according to mean residue evolutionary rate (ER) [25].

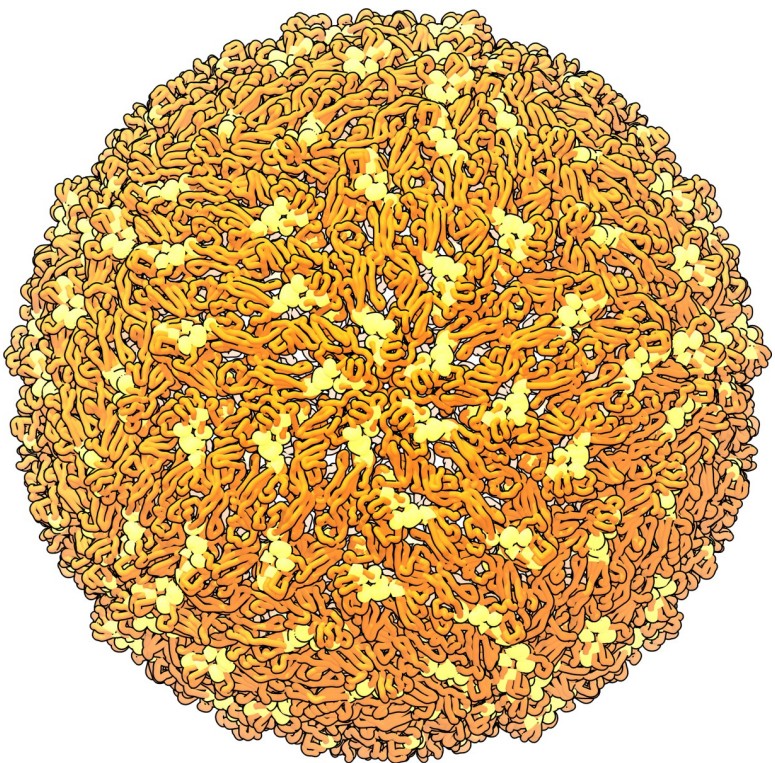

**Fig 4. Conserved epitope prediction demonstration.** Predicted conserved epitopes are represented in yellow on the ZIKV cryo-EM reconstruction backbone (red) (PDB ID: 5IRE) [8]. The cryo-EM reconstruction is shown with the five-fold rotational icosahedral symmetry axis coming out of the figure plane.

Comparison of the similarity [57] between full, predicted epitopes (clustered from predicted epitope residues) and structurally characterized epitopes shows several epitope predictions overlap with structurally characterized epitopes which were (not necessarily within the top seven conserved epitopes (as measured via ConSurf) [25] but still) bound by cross-reactive, neutralizing antibodies. Predicted epitopes with the highest similarity to structurally characterized epitopes were: [311, 312, 313, 314, 315, 316, 317, 318, 331] and [72, 73, 74, 75, 76, 97, 98, 99, 100, 101, 102, 103, 104, 105, 106, 107, 108, 109, 110, 111, 112]. Predicted epitope [311, 312, 313, 314, 315, 316, 317, 318, 331] showed 50%, 47%, and 45% similarity with epitopes bound by antibody 4E11 (PDB ID: 3UZE, 3UZQ, and 3UYP respectively). Antibody 4E11 cross-neutralizes all four dengue serotypes [58]. Predicted epitope [72, 73, 74, 75, 76, 97, 98, 99, 100, 101, 102, 103, 104, 105, 106, 107, 108, 109, 110, 111, 112] showed 36% similarity with an epitope bound by antibody 2A10G6 (PDB ID: 5JHL). Antibody 2A10G6 cross-neutralizes both ZIKV and dengue virus [59]. Predicted epitope [72, 73, 74, 75, 76, 97, 98, 99, 100, 101, 102, 103, 104, 105, 106, 107, 108, 109, 110, 111, 112] also overlaps with a structurally uncharacterized epitope bound by antibody 1C19 –which also cross-neutralizes all four dengue serotypes [60].

## Methods

### Model preparation and molecular dynamics simulations

ZIKV chains A, B, C, and E were extracted from a 3.80 Å resolution, cryogenic electron microscopy (cryo-EM) structure (Protein Databank (PDB) ID: 5IRE) [8]. Protein coordinates were run through Pulchra [61] to reconstruct missing and sterically clashing atoms. Protein

coordinates from chain A (residues 1–402) were used to construct a soluble, isolated protein simulation starting structure. An analogous process was applied to construct soluble, isolated protein simulation starting structures for six related flaviviruses (PDB: ID 4CCT (dengue virus serotype 1) [62], 3J27 (dengue 2) [63], 3J6S (dengue 3) [64], 4CBF (dengue 4) [65], 2HG0 (West Nile virus) [66], and 5WSN (Japanese encephalitis virus) [67]). For comparison, protein coordinates from ZIKV chain A (residues 1–502) and two chain Bs (touching chain A after application of the associated symmetry matrix) were used to construct a partially isolated ZIKV protein simulation starting structure (with transmembrane region and associated M proteins). Protein coordinates from ZIKV chains A, C, and E (residues 1–402) and an associated symmetry matrix (taken from PDB ID: 5IRE) were used to build a soluble ZIKV VLP simulation starting structure.

Using GROMACS 5.0.4 [68], explicit-solvent (TIP3P) [69] MD simulations were run on the flavivirus proteins and ZIKV VLP to a local structural equilibrium on the Indiana University Big Red II and III supercomputers (Fig 1). Attainment of local structural equilibrium was assessed as a steady time-average in backbone atom root-mean-square deviation (RMSD) relative to the starting configuration [20]. The CHARMM27 [70] forcefield was used with a 1 fs time step and periodic boundary conditions. The LINCS [71] algorithm was used to fix hydrogen bond lengths and the distance from protein to the edge of the cubic box was set to 1.5 nm. Steepest decent energy minimization was employed before simulation runs.

Flavivirus proteins were simulated in 142 mM NaCl (physiological) water baths. Simulation temperatures were coupled to 300 K (room temperature) using the Nosé-Hoover [72] thermostat. The coupling time constant was set to 0.5 ps, and ten Nosé-Hoover chains were used. The pressure was coupled to 1 bar using the Parrinello-Rahman [73] barostat. The coupling time constant and compressibility were set to 1 ps and 4.5E-5 bar. Short-range electrostatic cutoffs were set to 1.4 nm; long-range electrostatics were managed using the reaction-field [74] method. Epsilon-r (the relative dielectric constant) and epsilon-rf (the dielectric constant beyond a 1.4 nm reaction sphere cutoff) values were set to 1 (protein) and 78 (water).

Six cysteine residues pairs (([3, 30], [60, 121], [74, 105], [92, 116], [190, 291], and [308, 339]) which had sulfurs positioned 6 Å or less apart (overall mean sulfur separation distance of $3.5 \pm 1.2$ Å) on the ZIKV starting structures were simulated as disulfide bonds [8]. Disulfides, which aligned with those simulated for ZIKV, were also simulated for dengue virus serotype 1–4, West Nile virus, and Japanese encephalitis virus isolated protein starting structures. For comparison, the isolated ZIKV protein was also simulated without disulfides.

For simulations, the protonation state of titratable residues was chosen using a pH value of 8 (the pH associated with PDB ID: 5IRE) [8]. To identify biologically relevant pKa values for titratable flavivirus protein residues, a four-step procedure was performed as follows:

Step 1. Initial pKa guesses were estimated with PROPKA3.1 [75] using the starting model.

Step 2. Protonation states were assigned to titratable residues and a simulation was started.

Step 3. pKa values were predicted again using simulated protein coordinates averaged over an early portion of the trajectory. The simulation was continued if the pKa predictions matched pKas chosen in Step 1. Steps 2–3 were repeated with the new pKa values if predictions differed from pKas chosen in Step 1.

Step 4. pKa values were predicted again using simulated protein coordinates averaged over an equilibrated portion of the trajectory. Step 2–4 were repeated with the new pKa values if predictions differed from pKas found in Step 3.

An analogous process was performed for HPV simulation. HPV L1 protein chain A (PDB ID: 5KEP) [76] was used to construct the isolated HPV protein model. HPV L1 protein chains A–F (PDB ID: 3J6R) [77] were used construct the HPV VLP model. No protein truncation was performed, and no disulfides were simulated.

## Protein flexibility

Protein flexibility, measured as root-mean-square fluctuation (RMSF), was considered as an epitope discovery metric. Flavivirus and HPV backbone atom coordinate RMSF were calculated over structurally equilibrated 5 ns MD trajectory segments using GROMACS [68] as follows:

$$RMSF = \sqrt{\frac{1}{N}\sum_{n=1}^{N}\frac{1}{T}\sum_{t=1}^{T}|\vec{r}_n(t) - \langle\vec{r}_n\rangle|^2} \tag{1}$$

where $\vec{r}_n(t)$ is the position of backbone atom n of a residue with N backbone atoms at time t and $\langle\vec{r}_n\rangle$ is the time average position of atom n and T is the total number of discrete time steps. N, Cα and C atoms were included as backbone atoms. Rotation and translation of the isolated flavivirus proteins, the soluble portion (residues 1–402) of the partially isolated ZIKV protein, and each ZIKV VLP protein were subtracted before RMSF calculation. For the VLP, mean RMSF amplitudes were calculated for each residue over all proteins to enhance statistical significance, as all proteins which comprise the VLP are quasi-equivalent [78]. An analogous process was performed to quantify HPV RMSF.

## Layered convex hull depths

Solvent accessible surface area of the ZIKV isolated protein and VLP proteins were analyzed using a series of layered convex hulls. A convex hull is the smallest convex polygon which encloses all of the points within a dataset [79]. Each protein residue, (1–501) from chain A, C, and E, was reduced to its center of mass to construct three isolated ZIKV protein models for isolated protein convex hull [80] analysis (S2 Fig). Each residue (for the soluble portion (residues 1–402) of chain A, C, and E) was reduced to its center of mass and run through a symmetry matrix taken from PDB ID: 5IRE to construct a soluble ZIKV VLP model for convex hull analysis (S3 Fig).

Convex hull scores were calculated using an extension of a method described by Zheng, W., et al. (2015) [81] as follows:

Step 1. The SciPy [82] ConvexHull module was run on all model coordinates.

Step 2. Model coordinates which comprised the convex hull were given a score and then removed from the remaining model coordinates.

Step 3. Steps 1–2 were repeated until no coordinates remained.

Step 4. Residues in the first hull were given a score of one. Residues in the last hull were given a score of zero. Residues in the remaining hulls were given an evenly spaced score between zero and one based on the associated iteration of Steps 1–2. The mean of the scores was calculated for each residue.

## Preexisting structural epitope discovery metrics

ElliPro [32], DiscoTope [33], BEpro [34], Epitopia [35], and cons-PPISP [36] structure-based epitope and peptide discovery algorithms were run using default settings on ZIKV chain A, C, and E individually (residues 1–501), and the mean of the results was calculated for each

residue. ElliPro, DiscoTope, BEpro, and cons-PPISP structure-based epitope and peptide discovery algorithms were run using default settings on HPV chain A (the Epitopia server was unavailable at the time of HPV benchmarking). ZIKV temperature factors of chain A, C, and E were obtained from PDB ID: 5IRE [8], the square root was taken for each value, and the mean was calculated for each residue. Sequence-based intrinsically disordered protein region prediction algorithm, IUPred2A [47,48], was run using default settings. Spearman correlations between metrics were calculated using the SciPy [82] spearmanr module. T-tests were performed using the SciPy ttest_ind module (with equal_var set to False).

## Linear combinations of metric pairs

Linear combinations of select pairs of flavivirus structural metrics were calculated using two methods.

The first linear combination method is an extension of a method described by Kringelum, J. V., et al. (2012) [83] and was performed as follows:

For a pair of metrics, difference linear combinations (DLC) were calculated as

$$DLC(r, \alpha) = \alpha FS(r) - (1 - \alpha)SS(r) \tag{2}$$

where FS(r) represents the residue amplitude of the first partner of the pair, SS(r) represents the residue amplitude of the second partner of the pair, and $\alpha$ represents a weight from a grid of weights ranging from zero to one in increments of 0.005.

The second linear combination method was performed using weights which minimized the root-mean-square difference between the individual metrics (as discussed in the *Conserved epitope discovery performance benchmarking and the flexibility-based model* section of the *Results*).

## Structurally characterized, ZIKV-aligned flavivirus epitopes and HPV epitopes

Structurally characterized flavivirus epitopes were quantified using an extension of a method described by Stave, J. W. and K. Lindpaintner (2013) [6] and aligned to the ZIKV structure as follows:

Step 1. All available ZIKV, dengue virus, West Nile virus, and Japanese encephalitis virus envelope protein-antibody structures [12,50,58,59,84–101] were compiled from the National Institute of Allergy and Infectious Diseases Immune Epitope Database [102] (IEDB) and the PDB.

Step 2. The pairwise distance between each envelope protein atom and each antibody atom was calculated for each antibody within each structure.

Step 3. Envelope protein residues which had at least one atom within 4 Å from an antibody atom were identified and compiled for each structure, providing one epitope for each structure.

Step 4. Epitope residues were renumbered according to a structural sequence alignment between ZIKV and related flaviviruses reported by Kostyuchenko, V. A., et al. (2016) [8].

No envelope protein atoms were found within 4 Å of an antibody for PDB ID: 4C2I [90].

For conserved epitope quantification: ZIKV residue evolutionary rates were estimated using the cryo-EM chain A with ConSurf [25] (default settings) and the mean amplitude was calculated over each epitope. The sets of two and seven epitopes with the lowest mean residue evolutionary rates [12,59,86,88] were compiled (Table 4).

An analogous process was performed for HPV epitope quantification. HPV epitopes were aligned to the sequence of L1 protein chain A (PDB ID: 5KEP). Two structurally characterized epitopes with low mean residue evolutionary rates (PDB IDs: 7CN2 [54], 3J8Z [55]) out of eight epitopes were compiled for benchmarking.

## Epitope discovery performance benchmarking and rationale for a new method

Because of the high structural similarity between ZIKV and other flaviviruses [8], epitopes of other flaviviruses were considered as benchmark epitopes for ZIKV.

Epitope discovery performance benchmarking was extended from methods described by Ponomarenko, J., et al. (2008) [32] and Kringelum, J. V., et al. (2012) [83]. Two limitations of previous epitope discovery performance benchmarking approaches were addressed here. One limitation is the use of receiver operating characteristic curves on imbalanced datasets (a higher proportion of non-epitope residues (negatives) to epitope residues (positives)), which can overestimate how well a metric performs at identifying epitope residues (positives) [29]. This was overcome by benchmarking performance with precision-recall area under the curve (PRAUC) [103] analysis in conjunction with receiver operating characteristic area under the curve (ROCAUC) [104] analysis. A second limitation of previous studies is 1) measuring performance using individual epitopes and 2) then calculating the mean of the results. This was overcome by 1) benchmarking performance on a compilation of all epitopes using five stratified train-test data splits and then 2) calculating the mean of the results.

To understand why using of receiver operating characteristic curves to analyze epitopes individually can overestimate how well a metric performs at identifying epitope residues, one has to look at 1) the size discrepancy between an epitope and a pathogen protein and 2) the meaning of ROCAUC:

First, the average ZIKV-aligned flavivirus epitope is only ~5% of the total number (402) of soluble envelope protein residues.

Second, ROCAUC quantifies, over all thresholds, the combined probability of finding an epitope residue (positive) above a threshold over all epitope residues (positives) (true positive rates) and the probability of finding a non-epitope residue (negative) below the same threshold over all non-epitope residues (negatives) (true negative rates) (Fig 5). A 0.50 ROCAUC indicates performance no better than random chance (the value of no discrimination) [28]. Although ROCAUC is a valuable benchmark, when the proportion of non-epitope residues (negatives) is more pronounced than the proportion of epitope residues (positives), the probability of finding a non-epitope residue (negative) below a threshold over all non-epitope residues (negatives) (true negative rate) holds more weight within the combined probability than the probability of finding an epitope residue (positive) above a threshold over all epitope residues (positives) (true positive rate). In this case, ROCAUC provides more information about metric capacity to rank non-epitope residues (negatives) below epitope residues (positives) (true negative rates) than metric capacity to rank epitope residues (positives) above non-epitope residues (negatives) (true positive rates), which can mask weak true positive rates [29].

Because flavivirus epitopes are much smaller than the isolated proteins, benchmarking on individual epitopes (as done in the development of other epitope discovery algorithms) [5,32] results in datasets with a larger proportion of non-epitope residues (negatives) to epitope residues (positives)–introducing bias.

To understand why the use of precision-recall curves can more transparently reveal how well a metric performs at identifying epitope residues (positives) than the use of receiver operating characteristic curves, one must look at the meaning of PRAUC. PRAUC quantifies, over

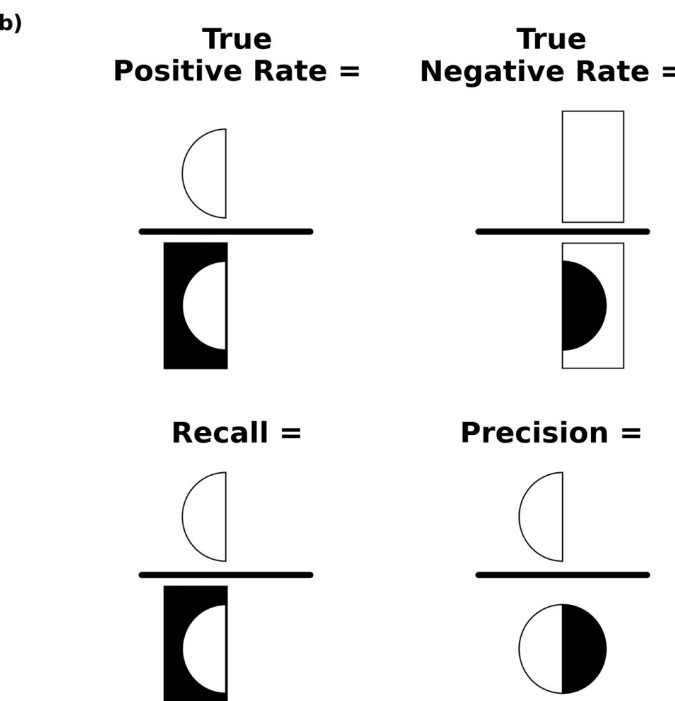

**Fig 5. ROCAUC and PRAUC benchmarking.** a) Possible prediction outcomes (used for ROCAUC and PRAUC benchmarking) (e.g. true positives) are depicted with black and white wedges. The vertical center line separates all positives and all negatives within the dataset. The center circle contains prediction outcomes for elements above a metric amplitude threshold. b) Rates (e.g. precision and recall) associated with possible prediction outcomes (used for ROCAUC and PRAUC benchmarking) are depicted with wedges and division bars.

all thresholds, the combined probability of finding an epitope residue (positive) above a threshold over all epitope residues (positives) (recalls or true positive rates) and the probability of finding an epitope residue (positive) above the same threshold over all residues (positives and negatives) also above the same threshold (precisions) (Fig 5). The proportion of epitope residues (positives) in the dataset indicates performance no better than random chance (the value of no discrimination) [28]. PRAUC is less biased than ROCAUC when there is a higher proportion of non-epitope residues (negatives) to epitope residues (positives) because, unlike ROCAUC, the probability of identifying non-epitope residues (negatives) below a threshold over all non-epitope residues (negatives) (true negative rate) is given no weight [29].

Epitope discovery benchmarking was performed as follows:

Step 1. In the first case, positives were defined as all structurally characterized, ZIKV-aligned flavivirus epitope residues and negatives were defined as all remaining flavivirus protein residues (from ZIKV envelope protein residues 1–402) (Table 1). In the second case, positives were defined as residues from the two and seven most conserved ZIKV-aligned flavivirus epitopes (Table 2).

Step 2. The epitope dataset was split into test (20%) and training (80%) datasets five times using the scikit-learn [105] StratifiedKFold module (n_splits = 5 and shuffle = True).

Step 3. For lone metrics, ROCAUC and PRAUC were calculated on the test datasets using the scikit-learn [105] roc_curve, precision_recall_curve, and auc modules and the means were taken.

Step 4. For linear combinations of metric pairs (as described in the *Linear Combination Metric Pairs* section above), ROCAUC and PRAUC were first calculated on the training datasets.

Step 5. The mean linear combination weights which produced the highest mean PRAUC and then the highest mean ROCAUC over the training datasets in Step 4 were identified.

Step 6. Using the optimized weights obtained in Step 5, ROCAUC and PRAUC were calculated against the associated test datasets and the means were taken.

An analogous process was used for HPV epitopes. Positives were defined as residues from two highly conserved HPV epitopes (PDB IDs: 7CN2 [54], 3J8Z [55]) out of eight epitopes. Negatives were defined as all other HPV residues in L1 protein chain A (PDB ID: 5KEP) (Table 3).

## Epitope organization and epitope discovery demonstration

After epitopic residues have been identified they must be connected to create whole epitopes–in the same way a collection of notes must be connected to form a song. Epitopes can be thought of as collections of subsequences along protein chains e.g. a continuous epitope has one subsequence and a discontinuous epitope with two parts has two subsequences. From this, each epitope was considered to have three major organizational properties: 1) a number of distinct subsequences, 2) a distribution of the number of residues in each subsequence, and 3) a number of total residues.

Experimentally characterized epitope subsequences were defined here through an extension of a definition described by Rubinstein, N. D., et al. (2008) and Sivalingam, G. N. and A. J. Shepherd (2012) [106,107] as one or more positives (including internal stretches of two or less negatives) bordered by stretches of three or more negatives on either side of the protein chain (when all epitopic residues were assumed to reside on an isolated protein). Predicted epitope subsequences were defined here as one or more residues above the prediction threshold

(including internal stretches of two or less residues below the prediction threshold) bordered by stretches of three or more residues below the prediction threshold on either side of the protein chain.

Epitope discovery was extended from methods described by Ponomarenko, J., et al. (2008) [32] and Kringelum, J. V., et al. (2012) [83] and performed as follows:

Step 1. For the top performing metric, a grid of metric amplitude thresholds (determined via the scikit-learn [105] precision_recall_curve module) and residue-residue distance cutoffs (1–10 Å in increments of 1 Å) was constructed.

Step 2. An amplitude threshold and residue-residue distance cutoff were chosen from the grid.

Step 3. Residues above the threshold chosen in Step 2 were clustered using the chosen residue-residue distance cutoff and a clustering method described by Ponomarenko, J., et al. (2008) [32].

Step 4. The mean number of distinct subsequences, the mean number of residues in each subsequence, and the mean number of total residues were calculated for residue clusters identified in Step 3.

Step 5. The mean number of distinct subsequences, the mean number of residues in each subsequence, and the mean number of total residues were calculated for experimentally characterized epitopes.

Step 6. The root-mean-square difference between quantities obtained in Step 4 and 5 was taken.

Step 7. Steps 2–6 were repeated until all grid points were exhausted.

Step 8. Residue clusters associated with the grid point which 1) minimized the root-mean-square difference calculated in Step 6 (sans those which contained residues 401–402 because the transmembrane region was not included in soluble protein simulations) and 2) contained five or more clusters were predicted as epitopes.

To compare the similarity of predicted epitopes with characterized epitopes, the Jaccard Similarity index [57] was calculated using residue numbering along the protein chain.

## Supporting information

**S1 Fig. Flavivirus isolated protein flexibility correlation plots.** Correlation plots for ZIKV isolated protein RMSF vs. dengue serotype 1 (DENV-1) isolated protein RMSF, dengue serotype 2 (DENV-2) isolated protein RMSF, dengue serotype 3 (DENV-3) isolated protein RMSF, dengue serotype 4 (DENV-4) isolated protein RMSF, West Nile virus (WNV) isolated protein RMSF, and Japanese encephalitis virus (JEV) isolated protein RMSF are shown. Isolated protein refers to a flavivirus envelope protein (with a transmembrane region truncation). Spearman rho (r) and p-values are also shown.
(TIF)

**S2 Fig. ZIKV isolated protein convex hull scoring.** The a) lowest scoring (most inner) and b) median scoring (middle) and c) highest scoring (most outer) isolated protein residue center of masses obtained from convex hull analysis are shown. Isolated protein refers to the ZIKV envelope protein.
(TIF)

**S3 Fig. ZIKV VLP protein convex hull scoring.** The a) lowest scoring (most inner) and b) median scoring (middle) and c) highest scoring (most outer) VLP protein residue center of masses obtained from convex hull analysis are shown. VLP protein refers to a ZIKV envelope protein (with a transmembrane region truncation) which comprises the VLP (a hollow protein cage).
(TIF)

**S1 Table. Epitope discovery performance benchmarking on individual epitopes.** Metrics are ordered from the top to bottom in terms of highest ROCAUC and PRAUC product. Spearman rho (r) and p-values are shown for associations between isolated protein convex hull scores (hmon) vs. Epitopia scores (opia), ElliPro scores (epro), isolated protein RMSF (fmon), VLP protein convex hull scores (hvlp), temperature factors (tfac), DiscoTope scores (dtope), VLP protein RMSF (fvlp), BEpro scores (bpro), cons-PPISP scores (ppisp), partially isolated protein RMSF (fmon*), and IUPred scores (iupred). The primary type of structural information utilized for each metric is shown under the heading titled Type (solvent accessible surface area (SASA), protein flexibility (RMSF), and sequence information (SEQ)).
(PDF)

**S2 Table. Flavivirus isolated protein flexibility: Conserved epitope discovery performance benchmarking.** Isolated protein flexibility of seven flavivirus structures is examined for epitope discovery performance against the top seven ZIKV-aligned, conserved flavivirus epitopes. Metrics are ordered from the top to bottom in terms of highest ROCAUC and PRAUC product. Spearman rho (r) and p-values are shown for associations between ZIKV isolated protein RMSF (zikv) vs. Japanese encephalitis virus isolated protein RMSF (jev), dengue serotype 2 isolated protein RMSF (denv2), West Nile virus isolated protein RMSF (wnv), dengue serotype 4 isolated protein RMSF (denv4), dengue serotype 3 isolated protein RMSF (denv3), and dengue serotype 1 isolated protein RMSF (denv1).
(PDF)

**S1 Dataset.**
(CSV)

## Author Contributions

**Conceptualization:** Daniel W. Biner, Jason S. Grosch, Peter J. Ortoleva.

**Data curation:** Daniel W. Biner.

**Formal analysis:** Daniel W. Biner.

**Investigation:** Daniel W. Biner.

**Methodology:** Daniel W. Biner, Jason S. Grosch, Peter J. Ortoleva.

**Validation:** Daniel W. Biner.

**Visualization:** Daniel W. Biner.

**Writing – original draft:** Daniel W. Biner, Jason S. Grosch, Peter J. Ortoleva.

**Writing – review & editing:** Daniel W. Biner.

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
