## [Decision Letter · Decision Letter 0]

29 Jun 2021

PONE-D-21-11419

B-cell epitope discovery: the first protein flexibility-based algorithm – Zika virus conserved epitope demonstration

PLOS ONE

Dear Dr. Ortoleva,

Thank you for submitting your manuscript to PLOS ONE. After careful consideration, we feel that it has merit but does not fully meet PLOS ONE’s publication criteria as it currently stands. Therefore, we invite you to submit a revised version of the manuscript that addresses the points raised during the review process.

A rebuttal letter that responds to each point raised by the  reviewers. You should upload this letter as a separate file labeled 'Response to Reviewers'.A marked-up copy of your manuscript that highlights changes made to the original version. You should upload this as a separate file labeled 'Revised Manuscript with Track Changes'.An unmarked version of your revised paper without tracked changes. You should upload this as a separate file labeled 'Manuscript'.

We look forward to receiving your revised manuscript.

Kind regards,

Paolo Carloni

Academic Editor

PLOS ONE

Journal Requirements:

Additional Editor Comments (if provided):

Reviewers' comments:

Reviewer's Responses to Questions

**Comments to the Author**

1. Is the manuscript technically sound, and do the data support the conclusions?

Reviewer #1: Yes

Reviewer #2: Yes

2. Has the statistical analysis been performed appropriately and rigorously? 

Reviewer #1: Yes

Reviewer #2: N/A

3. Have the authors made all data underlying the findings in their manuscript fully available?

Reviewer #1: Yes

Reviewer #2: No

4. Is the manuscript presented in an intelligible fashion and written in standard English?

Reviewer #1: Yes

Reviewer #2: Yes

5. Review Comments to the Author

Reviewer #1: The authors have carefully addressed the comments of the reviewers. Nonetheless, there are a few minor issues that the authors might want to address before the paper is published:

- The ZIKV protein simulations were performed without a membrane, even though the transmembrane region is included in the model. The authors explained very well what are the limitations of this set up in the response to the reviewers, but not in the manuscript. I would suggest that they add the same explanation in the main text, e.g. in the Methods ("Model preparation and molecular dynamics simulations" section).

- Maybe I overlooked it but I could not find the duration/length of the simulations. Can the authors indicate how many nanoseconds they ran for each system?

- The authors mentioned "cryptic epitopes", which exhibit low solvent accessible surface area in static structures. Have they investigated the time evolution of the SASA of these cryptic pockets along their MD trajectories?

- The authors explained in the response letter that they are negotiating with a repository to store their simulation files. For the sake of the data availability and reproducibility, the authors should include in the manuscript the link to the repository or offer the possibility to obtain the data upon request to the authors.

Reviewer #2: The authors present a very complete analysis of the epitopes on Zika virus surface. The techniques used are sound, and the simulations allow to improve the predictions of epitopes.

Although the work done is impressive, and considered that the authors improved a lot the manuscript after the revision, there are some elements that, from my point of view should be clarified:

1) I know that the systems are very big, but do the authors consider that just 50 ns MD simulations are enough to get stable conformations of the systems under study? Moreover, as the simulations are used to analyze the flexibility properties (RMSF, for example), Did the authors perform replicas in order to make the predictions more robust? These may strengthen all the conclusions.

2) I agree with the rational behind not using the explicit membrane in the ZIKV VLP system, but why not testing the putative effects of the membrane in the isolated systems, and then transfer this informartion to the ZIKV VLP system?

2) Moreover, the data are not available yet. Do the authors have updates regarding data availability. I am afraid that this is fundamental for publication in Plos One.

6. PLOS authors have the option to publish the peer review history of their article (what does this mean?). If published, this will include your full peer review and any attached files.

Reviewer #1: No

Reviewer #2: No

---

## [Author Response · Author response to Decision Letter 0]

24 Oct 2021

We have uploaded a response to the Reviewers.

---

## [Editor Report · Decision Letter 1]

23 Dec 2021

B-cell epitope discovery: the first protein flexibility-based algorithm – Zika virus conserved epitope demonstration

PONE-D-21-11419R1

Dear Dr. Ortoleva,

We’re pleased to inform you that your manuscript has been judged scientifically suitable for publication and will be formally accepted for publication once it meets all outstanding technical requirements.

Kind regards,

Paolo Carloni

Academic Editor

PLOS ONE
---

## [Editor Report · Acceptance letter]

27 Dec 2021

PONE-D-21-11419R1 

B-cell epitope discovery: the first protein flexibility-based algorithm – Zika virus conserved epitope demonstration 

Dear Dr. Ortoleva:

I'm pleased to inform you that your manuscript has been deemed suitable for publication in PLOS ONE. Congratulations! Your manuscript is now with our production department. 

Kind regards, 

on behalf of

Dr. Paolo Carloni 

Academic Editor

PLOS ONE